# Co-Adsorption of H_2_O, OH, and Cl on Aluminum and Intermetallic Surfaces and Its Effects on the Work Function Studied by DFT Calculations

**DOI:** 10.3390/molecules24234284

**Published:** 2019-11-25

**Authors:** Min Liu, Ying Jin, Jinshan Pan, Christofer Leygraf

**Affiliations:** 1National Center for Materials Service Safety, University of Science and Technology Beijing, Beijing 100083, China; min2@kth.se; 2Division of Surface and Corrosion Science, School of Chemistry, Biotechnology and Health, KTH Royal Institute of Technology, SE-100 44 Stockholm, Sweden; jinshanp@kth.se

**Keywords:** aqueous ad-layer, work function, micro-galvanic effect, DFT, aluminum, intermetallics

## Abstract

The energetics of adsorption of H_2_O layers and H_2_O layers partially replaced with OH or Cl on an Al(111) surface and on selected surfaces of intermetallic phases, Mg_2_Si and Al_2_Cu, was studied by first-principle calculations using the density function theory (DFT). The results show that H_2_O molecules tended to bind to all investigated surfaces with an adsorption energy in a relatively narrow range, between –0.8 eV and –0.5 eV, at increased water coverage. This can be explained by the dominant role of networks of hydrogen bonds at higher H_2_O coverage. On the basis of the work function, the calculated Volta potential data suggest that both intermetallic phases became less noble than Al(111); also, the Volta potential difference was larger than 1 V when the coverage of the Cl-containing ad-layer reached one monolayer. The energetics of H_2_O dissociation and substitution by Cl as well as the corresponding work function of each surface were also calculated. The increase in the work function of the Al(111) surface was attributed to the oxidation effect during H_2_O adsorption, whereas the decrease of the work function for the Mg_2_Si(111)–Si surface upon H_2_O adsorption was explained by atomic and electronic rearrangements in the presence of H_2_O and Cl.

## 1. Introduction

Al alloys are widely used in many applications because of their mechanical properties, easy fabrication, and good resistance to corrosion in various environments. While improving strength, intermetallic particles (IMPs) commonly increase the susceptibility of Al alloys to localized corrosion. Mg_2_Si and Al_2_Cu are among the most common phases in 2xxx and 7xxx aluminum alloys [1]. The coupling of these IMPs to an Al matrix results in the so-called micro-galvanic effects induced by their potential difference, which may trigger localized corrosion [2].

On one hand, Cl^−^ ions are believed to damage metal surfaces and hinder the re-passivation of localized corrosion of metals [3,4,5,6]. In a previous work, we studied co-adsorption of Cl and O_2_ molecules on an Al(111) surface by density functional theory (DFT) and found that the interaction of Cl and Al weakened the O–Al bond when the Cl/O ratio increased [5]. Moreover, co-adsorption of H_2_O and Cl^−^ ions on two of the common IMPs in Al, i.e., Al_2_Cu and Al_2_CuMg, was recently studied by DFT, and the results suggested that Cl^−^ ions can distort the structure within Al–Cu layers and lead to the formation of corrosion products [7]. On the other hand, surface adsorption was also frequently reported to alter the measured work function [8,9], which can consequently change the micro-galvanic effect. This issue was also addressed in a previous paper by the authors, suggesting that the adsorption of pure water had a significant effect on the relative nobility of different IMPs [10]. Thus, in order to further study the impact of a more corrosive environment on the micro-galvanic effect, Cl-containing aqueous ad-layers were explored in this work to possibly reveal their effect on micro-galvanic corrosion. 

The work function is the minimum energy needed to emit an electron from the surface to vacuum, implying that a higher work function corresponds to a higher resistance against losing electrons, and vice versa. This means that the work function can be regarded as an indicator of corrosion tendency [11,12]. Scanning Kelvin probe force microscopy (SKPFM) is usually applied to measure the Volta potential difference of two phases, which directly responds to the work function of the probe and the phases [13]. Hence, it is expected that factors that can affect the work function can also have an impact on the Volta potential difference.

Halogen adsorption was reported by DFT to decrease the metal work function. Examples include Cl on the (001) surface of Pt, Pd, and Rh [14], in which the polarization of halogen atoms overcompensates the charge transfer. Other studies have reported that Cl tends to increase the work function of Cu(111) [15] and Al(111) [16], while Br would raise the work function of Mg(0001) [17]. This can be explained by the strong electronegative character of halogen atoms, particularly Cl, which facilitates the formation of a dipole moment pointing from the halogen species to the metal surface [15]. A recent paper by Marks [18] reported that the galvanic effect between different sites on the oxide might be caused by a Cl adsorption-induced work function increase on several oxide surfaces, including Al_2_O_3_.

Computational studies have shown that the adsorption of H_2_O decreases the work function of metal surfaces, such as Pt [19,20,21], Pd [22], and Cu [23]. This was attributed to electron transfer from H_2_O molecules to the interfacial region between the water layer and the metal surface because of the large polarizability of H_2_O [21], and different H_2_O adsorption orientations were believed to be the main reason for different directions of work function change [22]. A work function decrease caused by H_2_O adsorption was also observed in experimental studies [24,25,26]. Moreover, a correlation between the adsorption energy of H_2_O and the work function of the H_2_O-adsorbed Pt(111) surface could be discerned [20].

It should be mentioned that Cl atoms, rather than Cl^−^ ions, are used in many DFT studies because of the lack of a clear description for ions in the calculation model, and some efforts have been made conceptually from an energetic point of view [27,28], in which the free energy of Cl^−^ was replaced with that of gaseous Cl_2_. On the other hand, DFT calculations can reveal whether charge transfer occurs or not, so the use of Cl atoms in the DFT study can still provide valuable insights in the adsorption process.

In an experimental work, the addition of pure Br_2_ molecules was found to increase the work function of Cu(110). Subsequent introduction of H_2_O, on the other hand, substantially decreased the work function, mainly because of the H_2_O dipole orientation [29]. With the development of computational capacity, theoretical efforts have been accomplished to approach more realistic solvation environments at the electrochemical liquid/solid interface by including explicit H_2_O molecules. For example, Wasileski et al. [30] concluded that a mild aqueous environment containing O_2_, Na, and H_2_O can reduce the work function fluctuations of Pt(111). 

The theoretical studies mentioned above mainly focused either on single adsorbing species or on non-aggressive aqueous ad-layers. To better understand the localized corrosion mechanism of Al alloys, efforts are needed to study a more corrosive aqueous environment (e.g., with both Cl and H_2_O) on both pure and hydroxylated Al and on relevant IMPs, which can mimic more realistic electrochemical corrosion conditions. The objective of this theoretical work is to study several aqueous environments relevant to corrosion scenarios by DFT calculations of the energetics of adsorption of H_2_O molecules and of Cl on Al(111) and on two IMPs, Mg_2_Si and Al_2_Cu, and also the subsequent H_2_O dissociation and substitution by Cl. The corresponding work function changes will be discussed from a galvanic corrosion perspective, aiming at shedding some light on the localized corrosion mechanism. This work is a continuation of our previous study [10], in which micro-galvanic corrosion between an Al matrix and Mg_2_Si or Al_2_Cu was explored both computationally and experimentally in the presence of a pure H_2_O ad-layers without co-adsorption of other species.

## 2. Computational Details

### 2.1. Model Construction

In this work, Al, Mg_2_Si, and Al_2_Cu crystals were considered. Aluminum has a cubic structure with a = 4.05 Å [31], and Mg_2_Si is cubic with a = 6.35 Å, while Al_2_Cu is tetragonal with a = 6.06 Å, c = 4.87 Å [32]. For Al, its (111) surface was studied. The Mg_2_Si(111) surface with Si and Mg terminations was selected, as these two terminations show rather different work function values. In addition, the Al_2_Cu(110)–Cu was selected for comparison. Note that vacuum was set to be at least as high as the thickness of the slab layers, and all atom positions were allowed to be optimized (details of the model construction can be found in [10,33]). In the following, Al(111), Mg_2_Si(111)–Si, Mg_2_Si(111)–Mg, and Al_2_Cu(110)–Cu will be simply written as Al, Mg_2_Si–Si, Mg_2_Si–Mg and Al_2_Cu–Cu, respectively.

To begin with, intact H_2_O adsorption (this configuration is shortened INT herein) on each metallic surface with different coverages was constructed. Firstly, one H_2_O molecule was placed on the top site of each surface with flat or vertical orientation. After optimization, the configuration with the lowest total energy was selected and duplicated on another top site. Since there were a few top sites on each surface, non-equivalent adsorption configurations were constructed and optimized, and only that with the lowest total energy was used for further analysis. In this work, only the top site was considered, focusing on ad-layer coverage.

On the basis of the final H_2_O adsorption configurations on each metallic surface, aqueous ad-layers including one hydroxyl group (OH) or one Cl atom, were subsequently constructed by one dissociated H_2_O (shortened DIS) and one substitution (SUB) of the dissociated OH by one Cl. To build the dissociation system, one H_2_O molecule was torn apart in the calculations, with the OH group remaining, and the H atom relocated to another top site. Hence, a few different dissociation ways were possible, and only those with the lowest total energy were selected and discussed. Figure 1 schematically depicts how the OH- and Cl-containing ad-layers were constructed. 

In this work, all results were discussed with the same coverage. The coverage of adsorbate, θ, is defined as the ratio of the number of adsorbates (e.g., water molecules, n_W_) to the number of atoms in the first layer. Here, coverages with 0.25 monolayers (ML), 0.5 ML, 1 ML, and 2 ML were considered. Note that the dissociation and substitution systems consist of a mixture of adsorbates, 1OH + 1H + (n_W_-1) H_2_O (DIS models) or 1Cl + 1H + (n_W_-1) H_2_O (SUB models). In the following, INT-θ, DIS-θ, and SUB-θ were used when needed. H_2_O orientations were not considered, as the focus was to create Cl-containing aqueous ad-layers.

Periodic DFT calculations together with the exchange–correlation functional, GGA-PW91, were performed using the Dmol3 code [34,35] implemented in MaterialsStudio. Core electrons were treated with DFT semi-core pseudopotentials (DSPP) with double numeric basis sets and polarization functions (DNP) [34]. A 6 × 6 × 1 k-point was used in all surface structures except for Al(111), for which a 3 × 3 × 1 k-point was used. Dipole correction was introduced vertically to each metallic surface to avoid dipole interactions. All atoms were relaxed until the energy, residual force, and displacement of each atom were less than 10^−5^ Ha (Hartree), 0.002 Ha/Å (Ångström), and 0.005 Å, respectively.

### 2.2. Description of Energetics

The work function of any surface configuration can be obtained by the energy difference between the vacuum level and the Fermi level, which is straightforward in DMol3 [33]. As proposed before, the Volta potential between different IMPs and the Al matrix is equal to their work function difference divided by e [33]. 

The adsorption energy per H_2_O, $E_{\mathrm{ad}}$, was calculated by Equation (1): (1)$$E_{\mathrm{ad}}\;=\;\left(E\left(\mathrm{Surface}+{\mathrm{nH}}_{2}O\right)-E\left(\mathrm{Bare}\;\mathrm{surface}\right)-n\cdot E\left(H_{2}O\right)\right)/n$$
where $E\left(\mathrm{Surface}+{\mathrm{nH}}_{2}O\right)$, E(Bare surface), and $E\left(H_{2}O\right)$ are the total energy of the H_2_O-adsorbed surface, bare surface, and H_2_O molecule, and n is the number of adsorbates. $E_{\mathrm{ad}}$ ˂ 0 means that the adsorption is exothermic (favorable), and vice versa. 

The energy needed for the dissociation ($E_{d}$) of one H_2_O molecule into OH and H and for substitution ($E_{s}$) of one OH by one Cl was calculated by Equations (2) and (3) [36], respectively:(2)$$E_{d}\;=\;E_{\mathrm{DIS}}-E_{\mathrm{INT}}$$
(3)$$E_{s}\;=\;E_{\mathrm{SUB}}\;+\;E_{\mathrm{OH}}\;-\;E_{\mathrm{DIS}}\;-\;1/2E_{{\mathrm{Cl}}_{2}}$$
where $E_{\mathrm{INT}}$, $E_{\mathrm{DIS}}$, and $E_{\mathrm{SUB}}$ are the total energy for the INT, DIS, and SUB adsorption systems, while $E_{\mathrm{OH}}$ and $E_{{\mathrm{Cl}}_{2}}$ are the energy for free OH and Cl_2_, respectively. 

## 3. Results and Discussion

### 3.1. Adsorption of Pure H_2_O Ad-Layers

As proposed in our previous work, water may adsorb in different configurations on different surfaces depending on their intrinsic properties. Nobility inversion was observed by DFT as well as by SKPFM when considering the work function change against coverage of water ad-layers [10]. In that previous study, the effect of H_2_O with different coverages on the work function was considered. It was concluded that the derived Volta potential difference between IMPs and the Al matrix agreed well with the experimental observations.

In this work, however, the adsorption energy per H_2_O molecule with increasing H_2_O coverage, $\theta_{H_{2}O}$, on each surface was further calculated by equation (1) and plotted in Figure 2. The results show that H_2_O adsorption on the Mg_2_Si–Si surface was unfavorable at low coverage, but became increasingly favorable with increasing $\theta_{H_{2}O}$, as indicated by a drastic drop in $E_{\mathrm{ad}}$ from positive values to −0.4 eV with 1 ML of H_2_O. This can be explained by a change of balance between H_2_O–surface interaction and H_2_O-H_2_O interaction [37]. A similar trend was seen for H_2_O adsorption on Al_2_Cu–Cu, whereas H_2_O adsorption on Al and Mg_2_Si–Mg was favorable at all coverages, with only a slight change in $E_{\mathrm{ad}}$ with H_2_O coverage (Figure 2). This could indicate that the interaction between H_2_O and these two surfaces is a dominant factor, irrespective of H_2_O coverage. However, as will be shown in a later section, the hydrogen bonding between H_2_O molecules is a dominant factor, making the interaction between H_2_O and the substrate surface of minor importance. Besides, it can be seen in Figure 2 that $E_{\mathrm{ad}}$ of all surfaces fell into a rather narrow range, −0.8 eV to −0.5 eV, with increasing coverage.

To understand the trend in adsorption energy shown in Figure 2, the energy difference between two hydrogen-bonded H_2_O molecules, $E_{H_{2}O–H_{2}O}$, and two free H_2_O molecules was calculated to evaluate hydrogen bond strength ($E_{H–\mathrm{bond}}$). The molecules were not adsorbed on any surface during the calculation. Note that the van der Waals (VdW) force between H_2_O molecules was not considered, as it is not implemented in the DMol3 package:(4)$$E_{H–\mathrm{bond}}=E_{H_{2}O–H_{2}O}-2\times E_{H_{2}O}$$

The initial distance between the two H_2_O molecules ($d_{0}$) bonded by hydrogen bonding was allowed to decrease from 2.45 Å to 1.65 Å. After optimization, a nearly constant distance, $d_{1}$ = 1.91 Å, and a corresponding $E_{H–\mathrm{bond}}$= −0.25 eV, were obtained, agreeing well with the experimental and theoretical values found in the literature [20,38].

The calculated $E_{H–\mathrm{bond}}$ suggests an explanation for the converging trend of the adsorption energy of H_2_O at higher H_2_O coverage (see Figure 2). Assuming that each H_2_O molecule has saturated hydrogen bonds, i.e., four H-bonds, then each H_2_O possesses two H-bonds and contributes −0.50 eV to the adsorption of H_2_O. The strength of H-bonds can be enhanced [20] or weakened [39] after the interaction with the substrate. With possible surface interactions neglected, we can conclude that hydrogen bonding is predominant when relatively large amounts of H_2_O molecules are present.

On Mg_2_Si–Si, one of the four water molecules was found to spontaneously dissociate into OH and H (Figure 3) when H_2_O adsorption reached 1ML. Looking back at the adsorption energy curve on the Mg_2_Si–Si surface (Figure 2), it is evident that the dissociated OH can promote water adsorption at 1ML. Besides, the dissociated H atom binds to surface Si, as shown in Figure 3b. Water dissociation on this surface may happen as the Si–H bond is stronger than the O–H bond in H_2_O [40]. Spontaneous dissociation was not observed on the other surfaces investigated. DFT calculations revealed a large energy barrier for H_2_O dissociation on Al(111) [41].

By plotting the work function at different H_2_O coverages of Al, Mg_2_Si–Si, Mg_2_Si–Mg, and Al_2_Cu–Cu (published previously [10]) against the corresponding adsorption energy, some special features were found (Figure 4). For Mg_2_Si–Mg and Al_2_Cu–Cu, the work function and adsorption energy were linearly correlated with one deviating value at 2 ML for Al_2_Cu–Cu. A favorable adsorption at 2 ML may be attributed to the formation of a water network on the Al_2_Cu–Cu surface. The deviating work function of this adsorption system may be due to the large dipole of H_2_O adsorbed far from the surface. A positive relationship was also observed for Mg_2_Si–Si. This correlation indicates that the more stable the adsorption (more negative $E_{\mathrm{ad}}$), the lower the work function of the adsorption system, in accordance with other studies [20,21]. The Mulliken charge analysis in our work implies that electron transfer from Mg_2_Si–Mg to H_2_O increased from 0.64 e to 0.77 e with increasing H_2_O coverage. It seems that electron transfer fails to explain the work function decrease, in comparison with the bare surface (blue dashed bar in Figure 4). We noticed that water molecules adsorbed on Mg_2_Si–Mg in an upward direction on average (Appendix A), which could compensate the effect of electron transfer. Therefore, water orientation is regarded as the predominant factor affecting the work function of Mg_2_Si–Mg.

For Al, the work function was nearly independent of the adsorption energy (Figure 4). Water molecules were ~ 4 Å away from the Al surface when the coverage was beyond 1 ML. As a result, a large dipole was built between water and Al, determining the work function change. On the other hand, hydrogen bond formation denoted a strong interaction among water molecules, as more water molecules were present on Al. Previous calculations confirm that the H_2_O adsorption energy is mainly determined by the hydrogen bond strength. Therefore, it is reasonable to conclude that the work function is independent of the adsorption energy on Al. The optimized structures of pure H_2_O on the four metallic surfaces are listed in Appendix A for comparison. We noticed that intact water adsorption caused a large surface relaxation on the Mg_2_Si–Si surface, even when only one H_2_O molecule was present. On the other hand, only a slight relaxation occurred on the Al_2_Cu–Cu surface with the formation of a Cu–H_2_O bond, and nearly no relaxation was seen on the other two surfaces (see Appendix A). It should be stressed, however, that there are many parameters that may influence the work function, such as surface relaxation, charge relaxation, and orientation of adsorbed H_2_O molecules. It is beyond the scope of this paper to be able to explain the relation between work function and adsorption energy for every investigated surface.

The above correlation between adsorption energy and work function provides complementary information about the reactivity of adsorbates towards the investigated surfaces (Figure 4). Though there may be several key factors which determine both work function and adsorption energy, carefully designed simulations or experiments may further reveal their influence on these interfacial properties, as reported previously [21,42].

The possibility of H_2_O dissociation on each investigated surface is discussed next. Since OH and Cl were reported experimentally to compete on metal surfaces [4], the substitution of OH by Cl is also discussed with respect to the mechanism of initiation of localized corrosion.

### 3.2. H_2_O Dissociation and Cl Substitution of OH

Hydrolysis is common on aluminum surfaces exposed to a damp atmosphere or aqueous solution. The resulting hydroxyl species can generate a protective film on the aluminum surface as it may oxidize the aluminum surface, similar to oxygen species [43]. Therefore, the “dynamics” of the aqueous environment at the metal surface is of great interest for the metal performance in damp and aqueous environments.

The dissociation energy $\;(E_{d}$) is a measure of the tendency of H_2_O dissociation. On the basis of the INT-aqueous ad-layer, the $E_{d}$ of H_2_O on the four surfaces was calculated by equation (2) with increasing $\theta_{H_{2}O}$ from 0.25 ML to 1 ML. Four or five dissociation paths were calculated at each H_2_O coverage, with several different top sites present on each investigated surface. To reduce the computing time, only up to 1 ML coverage was considered in each case, which is sufficient to represent the system. The minimum H_2_O dissociation energy, $E_{d\;\min}$_,_ versus H_2_O coverage is shown in Figure 5. More information about the dissociation energy for different paths is provided in the Appendix A).

The dissociation energy of H_2_O on Al and on Al_2_Cu–Cu was close to zero, nearly independent of H_2_O coverage, suggesting that water was unlikely to dissociate under the current calculation conditions. In contrast, the tendency for H_2_O dissociation on Mg_2_Si–Si was strong, although it became less favorable as the H_2_O coverage increased, indicating that the formed OH might inhibit further water dissociation. This was possibly due to limited surface sites available for the dissociated species. $E_{d\;\min}$ on Mg_2_Si–Mg decreased monotonically with water coverage, and H_2_O dissociation became favorable when $\theta_{H_{2}O}$ was larger than 0.5 ML. This can be explained by electron transfer from Mg_2_Si–Mg to the water layer, as mentioned in the previous section. That is to say, the surface Mg atoms were oxidized, indicating that reaction (5) proceeded in the right direction. Then, reaction (6) was promoted by the electrons from (5), increasing the tendency of H_2_O dissociation.
Mg → Mg^2+^ + 2e^−^(5)
H_2_O + e^−^ → H + OH^−^(6)

On the basis of the dissociated ad-layers obtained above, ad-layers containing Cl were constructed by replacing the dissociated OH with one Cl on each investigated surface. The substitution energy ($E_{s}$) values with increasing H_2_O are is listed in Appendix A. The substitution of OH by Cl in most cases is endothermic ($E_{s}$ > 0), except for Al and Mg_2_Si–Si at $\theta_{H_{2}O}$ = 0.5 ML, which suggests that Cl can barely replace OH. One reason could be that Cl, upon substitution, disrupts the hydrogen bond networks between OH and H_2_O, which is energetically expensive [44]. Another possible reason is that Cl has a larger ionic radius than the OH group, making the substitution process unfavorable [45]. Cl substitution of OH was also found to be endothermic on NiO(111) when over 70% of OH species were replaced [6]. 

However, real metal surfaces possess different kinds of defects, and it is well known that Cl^−^ ions can induce localized corrosion of many metallic materials. It is believed that defects or an applied electrode potential may drive the substitution process, as verified by other DFT studies [6,46]. Moreover, competitive adsorption has been observed experimentally [4]. Obviously, more complicated models relevant for real metal surfaces are needed for further calculations. Though the substitution process seems to be energetically favorable in only selected cases based on the current models and calculating parameters, the effect of Cl-containing ad-layers on the work function of Al(111) and Mg_2_Si–Si surfaces is discussed below for all coverages.

### 3.3. Effect on Work Function of Cl in the Aqueous Ad-Layer

On the basis of the above optimized structures within the three aqueous environments (INT, DIS, and SUB configurations), the work function of the four metallic surfaces upon adsorption of each ad-layer was calculated as a function of H_2_O coverage from 0.25 ML to 1 ML, see Figure 6.

The adsorption of pure intact water reduced the work function, compared to the value 4.16 eV for bare Al, as indicated by the dashed line in Figure 6a. This was due to the slightly upward pointing of the water molecules (Appendix A), which formed an upward dipole. In comparison with intact water adsorption, dissociation most likely increased the work function, except at coverage of 1 ML. This has been observed also on other metal surfaces, with OH being an electron acceptor [24,38,47]. Introduction of Cl raised the Al work function even further, as a result of a significant change of water molecules’ orientation upon adsorption of 1 ML of aqueous ad-layer, as seen in Appendix A and Figure 7a. 

Another important point depicted in Figure 7a is that Cl stayed above the water layer. Thus, a direct interaction between Cl and the Al surface was largely screened by the water molecules. In agreement with experimental data, Cl^−^ ions were observed to stay in the outer sphere of an Al^3+^ ion when surrounded by Cl^−^ and H_2_O molecules [48].

Intact water also decreased the work function of Mg_2_Si–Si at all coverages (Figure 6b). Different from the Al surface mentioned above, H_2_O dissociation or Cl substitution did not increase the work function. Cl adsorbed instead below the water layer, enabling Cl to interact directly with the surface at 1 ML of aqueous ad-layer adsorption (Figure 7b). On the other hand, the adsorption of Cl too close to the surface may be insufficient to create a surface dipole [14,49]. For comparison, the optimized structures for Mg_2_Si–Mg and Al_2_Cu–Cu upon adsorption of aqueous ad-layers containing Cl are also shown in Appendix A.

When focusing only on the Cl-containing aqueous ad-layer, we found that it increased the work function of Al but decreased that of Mg_2_Si–Si with respect to their bare surfaces (see Figure 6). To illustrate this phenomenon, deformation electron density (DED) maps on both surfaces are displayed in Figure 7c–g, by taking SUB-1 ML as an example.

On the Al surface, there was electron transfer from the outmost Al surface to the aqueous layer (Figure 7c). As a result, a dipole pointing from the water layer to the Al surface was built, which explains the work function increase. Meanwhile, this electron transfer also implied that the Al surface was oxidized by the water ad-layer. In qualitative agreement with this, experimental data have shown that the surface potential of an Al matrix increased after immersion in NaCl solutions [50]. Similarly, the work function increased in an Al–Mg alloy was attributed to surface oxidation by the water ad-layer [51], whereas the work function increase of Cu(110) with subsequent introduction of Br_2_ and H_2_O was attributed to the water dipole formed according to other experimental observations [29].

From Figure 7d it is seen, however, that slight electron density rearrangements occurred within surface Al, which could partly compensate for the work function change caused by the oxidation effect mentioned above [52]. Comparing the scale bars of Figure 7c,d, the work function increase by oxidation was dominant over that caused by rearrangements within the Al surface. Due to its strong electron affinity, Cl gained a concentrated electron density atop of it, while it created an electron depletion area around it, indicated by the color contrast in Figure 7d. However, direct electron transfer from Cl to Al was absent. By slicing another atomic plane (Figure 7e), Cl was demonstrated to gain electrons from an adjacent H_2_O, indicating a possible electronic interaction between these two species.

The DED map of the Mg_2_Si–Si, sliced through Si–H and the metallic plane (Figure 7f), revealed strong atomic as well as electron rearrangements in the surface layer. The relaxation of the metallic surface may also contribute to the change in work function [49]. Electron transfer between Mg_2_Si–Si and the aqueous layer was missing, which was expected since a bare surface with large work function is more resistant to losing electrons [53]. For visualization, we marked the electron-depleted area with “+” (white), and the electron-rich area with “−” (grey). Then a net dipole pointing upwards was seen in the first metallic layer and the aqueous ad-layer (Figure 7g), resulting in a decreased work function.

To demonstrate the impact of the Cl-containing aqueous ad-layer on the relative nobility of IMP and Al under the current conditions, we calculated Volta potential differences between the IMPs and Al, i.e., $V_{\mathrm{IMP}}-V_{\mathrm{Al}}$, covered by a SUB aqueous ad-layer, as shown in Figure 8. The data in Figure 8 show how the galvanic effect between the IMP and Al surfaces may vary depending on the Cl/H_2_O ratio in the ad-layer. Note that in the SUB aqueous ad-layer cases, since the concept of H_2_O coverage was also used here, the real ad-layer contained 1Cl + 1H + (n_W_-1) H_2_O. At 1 ML coverage, Cl occupied one of the surface sites of the calculation system, corresponding to a Cl/H_2_O ratio of 1:3.

The Volta potential difference data in Figure 8 indicate that upon ad-layer adsorption at 0.25 ML, with actually the adsorption of Cl + H on the Mg_2_Si–Si surface and of Cl + H + H_2_O on the Al_2_Cu–Cu surface, the two IMPs transformed from cathode (bare) to anode relative to Al, while Mg_2_Si–Mg remained anodic. As the coverage increased to 0.5 ML, the Volta potential difference of all the three IMPs relative to Al increased in comparison with a decrease in coverage, and the two Mg_2_Si surfaces terminated by Mg and Si remained anodic relative to Al, while Al_2_Cu–Cu changed back to cathode at 0.5 ML. Furthermore, as the coverage of the Cl-containing ad-layer reached 1 ML, all three IMP surfaces became less noble than the Al surface, with a rather large Volta potential difference (>1 V). As can be seen from the work function at SUB-1 ML in Figure 6 and Appendix A, the Al work function was the highest among those of all the studied surfaces, mainly due to the strong oxidation phenomenon, as described in Figure 7c. 

Meanwhile, the initial galvanic couples Mg_2_Si–Si (cathode)/Al (anode) as well as Al_2_Cu–Cu (cathode)/Al (anode) were reversed with the adsorption of the SUB aqueous ad-layer, indicating a possible mechanism for the previously observed, so-called “nobility inversion” [54]. In all, this implies that different interactions (e.g., electron transfer and dipole formation) between the metallic surface and the aqueous ad-layer can result in changes of the galvanic effect. As has been recently calculated by L. Marks [18], Cl adsorption can increase the work function of Al_2_O_3_, which promotes heterogeneous oxidation in areas where Cl adsorbs, compared to areas where there is no Cl adsorption. 

### 3.4. Short Summary and Implications

In this theoretical work, we studied the effect of Cl-containing aqueous ad-layers on the Volta potential difference between three intermetallic surfaces and Al. This was accomplished by DFT calculations of the energetics of adsorption and dissociation of H_2_O as well as by substitution of OH with Cl within the ad-layers, followed by calculations of the work function of the surfaces covered by the aqueous ad-layers. To begin with, explicit H_2_O molecules up to 2 ML adsorbed on Al, Mg_2_Si–Si, Mg_2_Si–Mg, and Al_2_Cu–Cu surfaces were studied. Then, one OH was introduced by dissociation of one adsorbed H_2_O molecule, and afterward one Cl was introduced into the aqueous ad-layer by substitution of the OH. In this way, a more realistic corrosive environment was constructed for the assessment of the effect of the Cl-containing ad-layer on the galvanic corrosion due to the coupling between the IMPs and the Al matrix. These three model scenarios correspond to molecular H_2_O adsorption, subsequent H_2_O dissociation, and competitive adsorption between Cl and OH on an Al alloy surface containing Mg_2_Si or Al_2_Cu particles. 

The results showed that intact H_2_O adsorption on all four surfaces was exothermic when the H_2_O coverage increased to 1 ML (Figure 3), and an enhanced H-bonding within the H_2_O ad-layer was observed. The introduction of one OH in the intact H_2_O ad-layer showed that H_2_O dissociation was not favorable on Al and Al_2_Cu–Cu under the examined calculation conditions, whereas H_2_O had a strong tendency to dissociate on Mg_2_Si–Si and Mg_2_Si–Mg (Figure 5). By calculating the reaction free energy of H_2_O dissociation on the Al surface based on the methodology proposed by J.K. Nørskov [55], we found that H_2_O dissociation was favorable at a positive electrode potential (to be published elsewhere) [56]. In the future, the VdW corrections should also be taken into account because of the moderate modification of the adsorption energy produced by the VdW force, especially for energies around 0.1 eV [57].

When introducing one Cl into the aqueous ad-layer in replace of the OH, the present calculation indicated that the substitution process was endothermic in most of the calculation conditions (Appendix A). Since it is well known that chloride ions usually promote the corrosion of metals, further considerations are needed in the interpretation of the calculation results. Real metal surfaces contain different kinds of defects and are much more complicated than the calculation models used herein, which only represent single-crystal surfaces. On the other hand, DFT calculations, though only for simple systems, provide a fundamental understanding of the surface processes and interactions at the atomic scale. For example, when one Cl atom is added to the surface, DFT calculations can tell where electrons go and thus provide information about charge transfer or dipole formation, which can be viewed by the DED maps sliced at different atomic planes of the surface layer. Such information is helpful for understanding the effect of surface adsorbates on the work function of metal surfaces. The calculation of low-H_2_O-coverage models yielded some theoretical insights, while the model with one full monolayer (1 ML) of aqueous adsorbates was more relevant for corrosive environments.

The work function of the metallic surfaces with Cl together with H_2_O molecules under varying H_2_O coverages (Figure 6a,b) agrees with both theoretical and experimental studies [15,16,29]. On Al, the adsorbed H_2_O layer attracted electrons from Al surface atoms (Figure 7c), forming a dipole at the surface and causing an increase in the work function [29]. The effect of Cl, positioned nearly 5 Å above the surface, seemed to be screened by the H_2_O molecules. The DED map evidenced electronic interactions between Cl and adjacent H_2_O (Figure 7e). In contrast, the adsorption of SUB-1 ML aqueous ad-layer on Mg_2_Si–Si led to a serious rearrangement both structurally and electronically (Figure 7f). The upward dipole formed at the surface layer was the main reason for the work function decrease (Figure 7f,g). 

Derived from the work function data, the Volta potential difference between the three IMP surfaces and Al covered by the adsorbates suggests that co-adsorption of Cl and H_2_O molecules makes the IMPs significantly less noble compared to Al at 1 ML coverage of the aqueous ad-layer (Figure 8). This may provide an explanation for the effect of chloride ions in promoting localized corrosion of Al alloys. In this case, the dissolution of the IMPs is driven by micro-galvanic coupling with the Al matrix.

## 4. Conclusions

In this study, we constructed a series of aqueous ad-layers containing H_2_O molecules up to two monolayers and aqueous ad-layers with one OH group or Cl atom on Al(111), Mg_2_Si(111)–Si, Mg_2_Si(111)–Mg, and Al_2_Cu(110)–Cu surfaces, seeking to mimic a simplified but, yet, realistic system relevant for micro-galvanic corrosion of Al alloys containing Mg_2_Si or Al_2_Cu particles. DFT calculations were performed to obtain the energetics of adsorption of H_2_O ad-layers, during dissociation of H_2_O and subsequent substitution of OH by Cl, the work function of the surfaces covered by the different aqueous ad-layers, and Volta potential differences between the intermetallic surfaces and Al with a Cl-containing aqueous ad-layer. The following conclusions can be drawn:

For adsorption of pure H_2_O, there was an enhancement of H bonding under higher H_2_O coverage, which promoted stable adsorption on the surfaces and adsorption energies in a relatively narrow range between −0.8 and −0.5 eV for all surfaces investigated.

Spontaneous H_2_O dissociation was only observed on Mg_2_Si(111)–Si, where H was bound to surface Si atoms. A large dissociation tendency was seen on Mg_2_Si(111)–Si and also on Mg_2_Si(111)–Mg at higher H_2_O coverage. However, subsequent substitution of OH with Cl was largely prevented within explicit H_2_O ad-layers, especially at higher H_2_O coverage. 

Electron transfer from surface Al atoms to the H_2_O ad-layer can explain the work function increase of Al(111) covered by the Cl-containing aqueous ad-layer, whereas atomic and electronic rearrangements on Mg_2_Si(111)-Si are the main reason for the work function decrease caused by the Cl-containing aqueous ad-layer. 

The Volta potential difference between the intermetallic surfaces and Al(111) suggests that co-adsorption of Cl and H_2_O molecules makes the IMPs significantly less noble compared to Al(111) at 1ML coverage of the aqueous ad-layer.

## Figures and Tables

**Figure 1 molecules-24-04284-f001:**
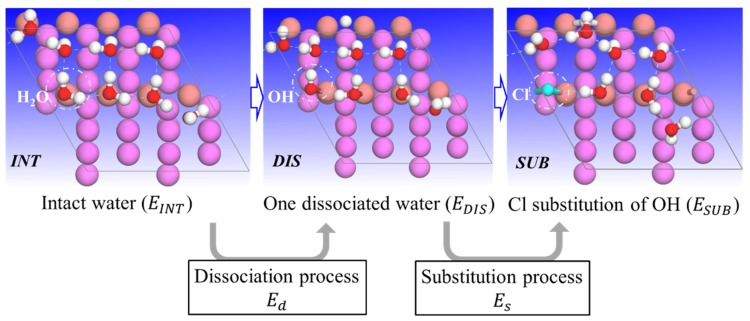
Top view depicting the different calculated configurations of adsorbed aqueous ad-layers: intact H_2_O (INT), water with one OH (DIS), and water with one Cl (SUB). The metallic surface for H_2_O adsorption is only a representative substrate. Note that in the SUB case, a surface bond is formed between the Cl and a surface atom.

**Figure 2 molecules-24-04284-f002:**
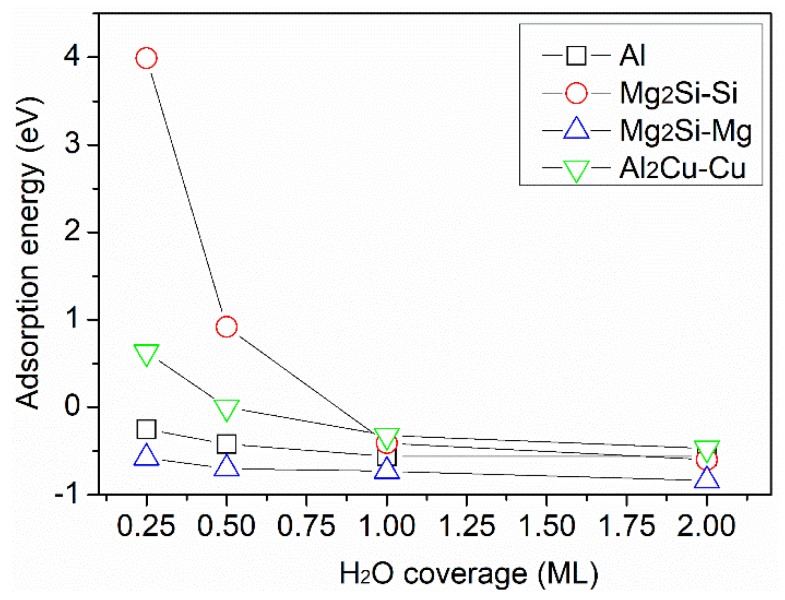
Adsorption energy of H_2_O on Al, Mg_2_Si–Si, Mg_2_Si–Mg, and Al_2_Cu–Cu surfaces versus H_2_O coverage.

**Figure 3 molecules-24-04284-f003:**
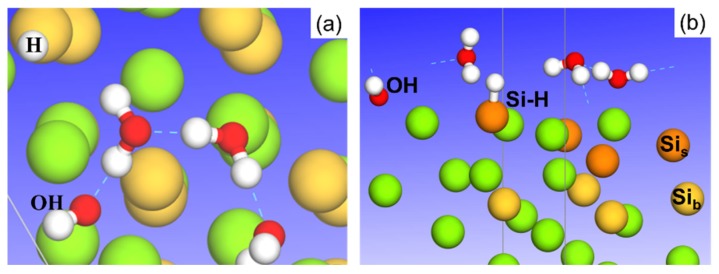
Optimized structure of H_2_O dissociation into OH and H (binding to surface Si) on Mg_2_Si–Si with 1ML H_2_O adsorption: (**a**) top view, (**b**) side view. Surface Si (Si_s_) and bulk Si (Si_b_) are indicated by dark and light orange color. Mg is in green, O in red, and H in white.

**Figure 4 molecules-24-04284-f004:**
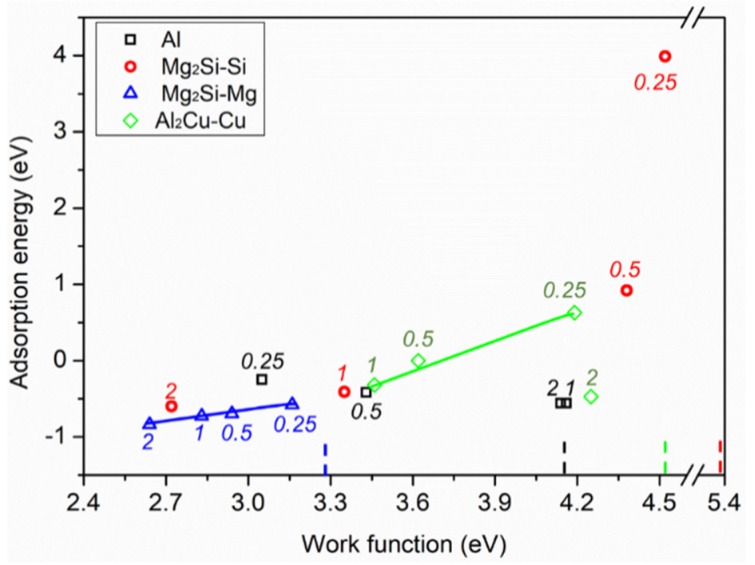
Plots of H_2_O adsorption energy on different surfaces versus work function under different H_2_O coverage (see the values without unit for each surface shown in the figure). The work function for each bare surface is highlighted by a dashed bar. Work function values for intact H_2_O adsorption were reported in our previous work [10].

**Figure 5 molecules-24-04284-f005:**
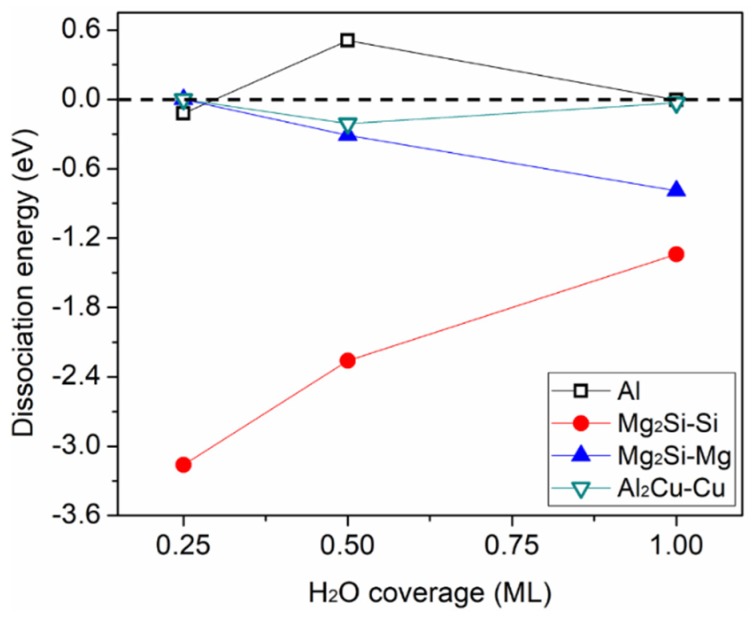
The minimum dissociation energy, $E_{d\;\min}$, as a function of H_2_O coverage for the four investigated surfaces.

**Figure 6 molecules-24-04284-f006:**
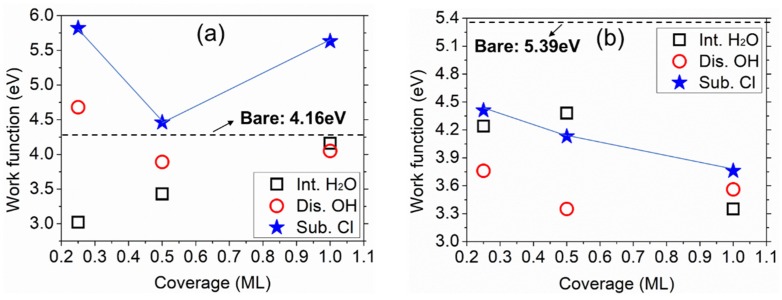
Work function change with adsorbed INT, DIS, SUB aqueous ad-layers on (**a**) Al and (**b**) Mg_2_Si–Si. Data for bare surfaces and INT aqueous ad-layer systems are from our earlier work [10,33].

**Figure 7 molecules-24-04284-f007:**
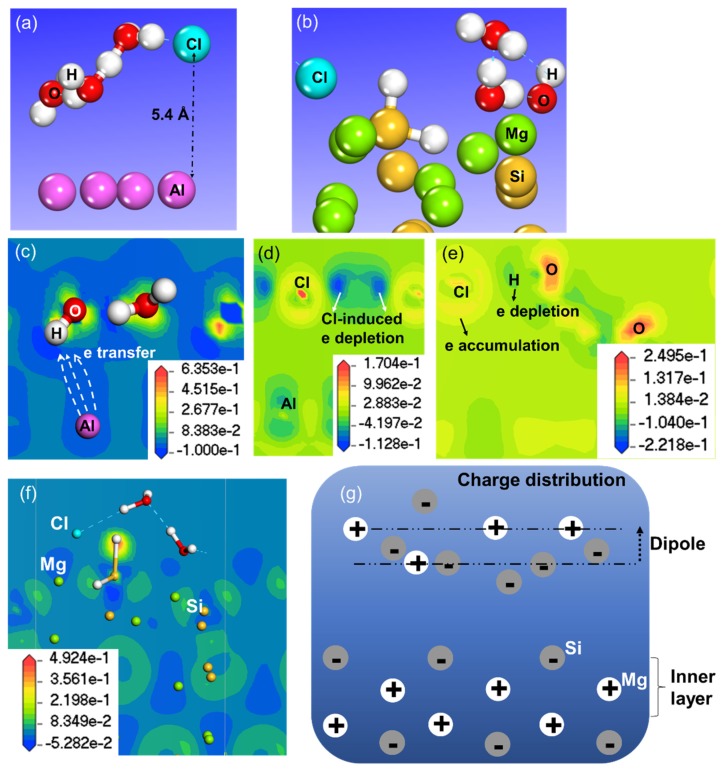
Optimized structure of Cl-containing aqueous ad-layer on Al (**a**) and Mg_2_Si–Si (**b**). Deformation electron density (DED) maps on Al sliced through (**c**) one H_2_O plane, (**d**) the Cl atom, (**e**) both the Cl atom and H_2_O. DED maps on Mg_2_Si–Si (**f**) and schematic charge distribution in the surface layers (**g**) based on Figure 7f. In Figure 7c–f, the blue and red colors indicate electron deficiency and accumulation, respectively.

**Figure 8 molecules-24-04284-f008:**
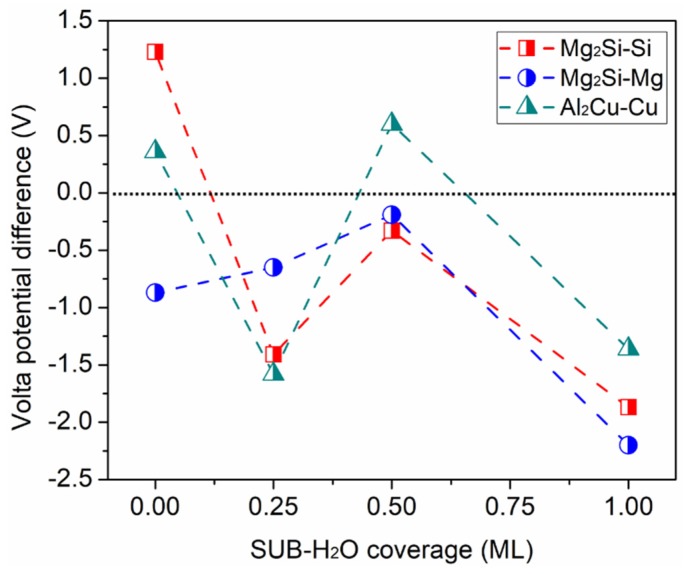
Volta potential difference of Mg_2_Si–Si, Mg_2_Si–Mg, and Al_2_Cu–Cu relative to Al, with the adsorption of a Cl-containing aqueous ad-layer at different coverage levels.

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
