# Peer review of "Co-Adsorption of H2O, OH, and Cl on Aluminum and Intermetallic Surfaces and Its Effects on the Work Function Studied by DFT Calculations"

_molecules, 2019, doi:10.3390/molecules24234284_

Round 1

Reviewer 1 Report

Minor Revisions

A revision is needed on the numbering system of Equations (1) and (2). The description of each data series in Figure 2 is a little difficult to read. Perhaps, the authors can include a box legend in the this figure (in the same way they did with Figure 4) In page 5, the estimation of the hydrogen bond strength between 2 water molecules is presented using Eq. 4. It is not very clear if the authors employed adsorbed water molecules or isolated water molecules to calculate the EH-bond value. In page 6 the authors wrote “For Mg2Si-Mg and Al2Cu-Cu, the work function and adsorption energy are linearly correlated with one deviating value at 2 ML for Al2Cu-Cu.” Can you provide a possible explanation for this deviation observed in the work function for Al2Cu-Cu. In page 8, line 279, the authors wrote “In comparison with the intact water adsorption, dissociation increases the work function” By looking Figure 6 it is observed that at ML = 1 this is not true. The work function of dissociation is slightly lower than the intact water adsorption function. Please, check this statement.

Reviewer 2 Report

The authors present an original interesting physico-chemical study, which deals with adsorption of halogens, water and hydroxyl group on surface of metals or selected intermetallic phases. 

The authors investigated and clarified especially the co-adsorption processes, which are important in material sciences. and can show a future impact for studies of corrosion.  In this context, the topic of the manuscript is timely.

The authors used adequate methods, which are sufficiently described. The data support the results, which are properly discussed. Additionally,  the conclusions are well presented and correspond with the presented results. Furthermore, I appreciate the supplementary material. The supplementary figures and table illustrate additionally the results. 

Since I found no serious mistakes in the manuscript, I recommend the paper for publication.

Author Response

Thanks for the reviewer' s positive opinions and recommendation for publication!

Reviewer 3 Report

Comments on Co-adsorption of H2O, OH, and Cl on aluminum and intermetallic surfaces…

The authors present DFT calculations on adsorption of water on the Al (111) surface and include intermetallic systems Mg2Si and Al2Cu. The propensity toward corrosion is described by the response of the systems to water dissociation and the presence of Cl. They explore specific cases INT (in which the water molecules are intact), DIS (in which one H2O is fragmented to H and OH), and SUB (in which the OH of DIS is replaced by a Cl). The issues of water orientation and dissociation pathways are addressed. The fraction of site coverage adsorbate species per metal site (theta) is varied from 0 (bare) and 0.25 to 1.0 or in some cases 2.0. The Dmol3 code implemented in MaterialsStudio is used to generate structures and energies. The adsorption energy per adsorbed water molecules (INT) vs. fraction of coverage approaches a common limit for high theta, suggesting a H-bonding water layer interacting weakly with the (inter)metal surface.

The relation between work function (energy required for detachment of an electron from the metal) is complex. For Al, low coverage (theta = 0.25) reduces the work function sharply though the adsorption energy is small. As coverage increases the work function approaches that of the bare surface. This is consistent with the picture of weakly interaction of the metal and the water monolayer. For Mg2Si:Mg  terminus and Mg2Si:Si terminus the work function decreases with increasing coverage (up to theta = 2), slowly for the former and sharply for the latter. The behavior for Al2Cu:Cu terminus is not so simple. The odd result at theta = 2 is noted but not explained. (See note 1 below.)  

The energy change attending water dissociation is dependent on theta. For Mg2Si:Si the first adsorption is strongly exoergic (-3 at theta = 0.25) whereas the Mg2Si:Mg is almost thermoneutral. The dissociation energy for the first system weakens as coverage increases, while the dissociation energy for the second system becomes more exoergic for high coverage. (See note 2 below.)

The work function depends on coverage as well, in rather complicated ways. Two examples are provided. For Al, the work function is strongly reduced for low theta (we assume that water interacts with an Al in this limit) but has little impact at high coverage (we assume the H-bonded monolayer is established, and interacts weakly with the metal surface.) DIS and SUB systems show an elevation of the work function at small theta and high theta, with a lesser impact at intermediate theta. The SUB system shows greater work function elevation than does DIS, arguing for stronger interaction for Cl than OH with the Al surface. For the Mg2Si:Si terminus system, INT, DIS, and SUB systems all show reduction in work function. (See note 3.) The authors present differential electron density diagrams to explain the Cl system behavior.

The voltage difference V(intermetallic) – V (Al) differs for the Si terminated and Mg terminated systems ((+1.25 for Si, -0.8 for Mg) and converge to about -2eV only at high coverage (theta = 1). Their behavior is more or less parallel for theta at 0.25 and greater values. (See note 4.)

The authors have chosen an interesting set of model systems, and address an important issue, the connections among composition, coverage and corrosion in pure metal and intermetallic systems. The presentation is clear, apart from one issue. I wonder why the Mg terminated and Si terminated systems behave so differently, and how the choice of terminator is related to an experimental situation.

I also feel the absence of any discussion of implications for design and any generalizations on what might be expected for other fairly similar systems. The conclusions section is a compact summary of presented information; a broader discussion of possible extensions would be welcome.

Note 1: The authors may wish to examine the structure of the Al2Cu:Cu terminus system at theta = 2 to see whether a structural change has occurred.

Note 2: The choice of terminus seems to have a striking impact on water dissociation energy. I wonder if there is a structural explanation?

Note 3: Since there seems to be a significant difference between Mg and Si terminated systems, shouldn’t we see data for the Mg terminated system?

Note 4: Why the big effect for low coverage?

Round 2

Reviewer 4 Report

Title: Co-adsorption of H2O, OH and Cl on aluminum and intermetallic surfaces and its effects on work function studied by DFT calculations

Authors: Min Liu, Ying Jin, Jinshan Pan and Christofer Leygraf

SECOND REVIEW

The authors revised their paper in accordance to part of reviewer’s comments. But some responses are still missing.

Comments and questions about the paper:

5) Adsorption of pure H2O ad-layers:

i) First review: “ The adsorption energy shows that the interactions between the H2O molecules and the substrate are stronger with increasing coverage up to 1ML. For Mg2Si-Si especially it becomes favorable for 1ML: the authors conclude that there is a “balance between H2O-surface interaction and H2O-H2O interaction” (line 163) whereas for Al(111) “the interaction between H2O and these two surfaces is a dominating factor” (line165). This could be illustrated by the decomposition of the adsorption energy into (i) the H2O/H2O interaction in the ad-layer (cohesive energy of the layer for instance) and (ii) the surface/H2O layer interaction (calculating the difference between the energy of the system and the energy of the isolated slab and H2O layer at the geometry after adsorption).”

 This comment has not been taken into account by the authors.

The reviewer underlines that the authors would better characterize the interactions within the system if they illustrate what they wrote “the balance between H2O-surface interaction and H2O-H2O interaction” and “the interaction between H2O and these two surfaces is a dominating factor” with energy values by calculating the H2O-surface interaction energy and interaction energy of the H2O molecules in the H2O layer. It could be expected that interaction of the metal and the water monolayer is weak.

ii) First review: On the Figure 4, the authors propose that there is a linear correlation between the adsorption energy and the work function. Again, the charge transfers would help to explain the work function changes. It seems that there is no direct correlation between the work function and the value of the coverage Al: WF 0.25<0.5<2<1 & Al2Cu-Cu WF: 1<0.5<0.25<2 for instance). (Why?

Responses from the authors: Thanks for commenting the deviation on Al2Cu surface at 2ML. Water molecules adsorb in an exothermic way (1), while the work function of this system attains, not lower than that at 1ML (2) as expected.

That is, there is a deviation of Al2Cu at 2ML, Reason for (1) might be the formation of “hydrogen bonding network”, which contributes to the adsorption energy, as has been stated in section 3.1. Meanwhile, water adsorption leads to electronic relaxation around the surface copper atoms from the electron density analysis, resulting in a slight increase of work function (this is the reason for (2)). In contrast, surface relaxation is not observed in the 1ML case.

The reviewer does not only comment the deviation on Al2Cu surface at 2ML. For Al2Cu-Cu and Mg2-Si systems, the WF decreases as the coverage increases. For Al, the WF increases when the coverage increases. The authors could try to explain these different cases.

7) First review :  Effect on work function of Cl in the aqueous ad-layer: - No information is given in the SI about the geometries when Cl substitutes OH (line 282). Is the case 1 ML given in Figure 7a?

Responses from the authors: Thanks for the comments. In this manuscript, the focus is on the Cl-containing aqueous ad-layer, thus only configurations with Cl-are shown in the main part of the manuscript.

What is the coverage in figure 7a? What about the topology of the Cl-containing aqueous ad-layer for other coverages or substrate that are not shown?
